# Predictive Roles of Basal Metabolic Rate and Muscle Mass in Lung Function among Patients with Obese Asthma: A Prospective Cohort Study

**DOI:** 10.3390/nu16121809

**Published:** 2024-06-08

**Authors:** Xin Zhang, Li Zhang, Ying Liu, Lei Liu, Ji Wang, Changyong Wang, Shuwen Zhang, Gaiping Cheng, Lei Wang

**Affiliations:** 1Division of Internal Medicine, Institute of Integrated Traditional Chinese and Western Medicine, West China Hospital, Sichuan University, Chengdu 610041, China; zhangxinwch@126.com (X.Z.);; 2Department of Respiratory and Critical Care Medicine, Clinical Research Center for Respiratory Disease, West China Hospital, Sichuan University, Chengdu 610041, China; 3Department of Clinical Nutrition, West China Hospital, Sichuan University, Chengdu 610041, China

**Keywords:** obese asthma, basal metabolic rate, muscle mass, lung function

## Abstract

Background: The metabolic-status-related mechanisms underlying the deterioration of the lung function in obese asthma have not been completely elucidated. Objective: This study aimed to investigate the basal metabolic rate (BMR) in patients with obese asthma, its association with the lung function, and its mediating role in the impact of obesity on the lung function. Methods: A 12-month prospective cohort study (n = 598) was conducted in a real-world setting, comparing clinical, body composition, BMR, and lung function data between patients with obese (n = 282) and non-obese (n = 316) asthma. Path model mediation analyses for the BMR and skeletal muscle mass (SMM) were conducted. We also explored the effects of the BMR on the long-term lung function in patients with asthma. Results: Patients with obese asthma exhibited greater airway obstruction, with lower FEV_1_ (1.99 vs. 2.29 L), FVC (3.02 vs. 3.33 L), and FEV_1_/FVC (65.5 vs. 68.2%) values compared to patients with non-obese asthma. The patients with obese asthma also had higher BMRs (1284.27 vs. 1210.08 kcal/d) and SMM (23.53 vs. 22.10 kg). Both the BMR and SMM mediated the relationship between obesity and the lung function spirometers (FEV_1_, %FEV_1_, FVC, %FVC, and FEV_1_/FVC). A higher BMR or SMM was associated with better long-term lung function. Conclusions: Our study highlights the significance of the BMR and SMM in mediating the relationship between obesity and spirometry in patients with asthma, and in determining the long-term lung function. Interventions for obese asthma should focus not only on reducing adiposity but also on maintaining a high BMR.

## 1. Introduction

The complex interplay between obesity and asthma poses a significant challenge in understanding the pathophysiological mechanisms underlying this comorbidity [1]. Obesity, as a primary risk factor, exacerbates asthma symptoms and the disease severity, giving rise to the phenotype known as “obese asthma.” This particular asthma subtype is often unresponsive to standard therapeutic approaches [2], emphasizing the need for more tailored treatment strategies. Abnormal lung function, particularly in patients with obese asthma, serves as a strong predictor for uncontrolled asthma and asthma exacerbations [2]. Decades of research have focused on elucidating the physiological effects of obesity on the lung function [3], yet the association between asthma and obesity has sparked renewed interest in exploring the mechanical impacts of obesity on the lung function. These mechanisms remain controversial and not fully understood [4]. One such mechanism involves the impact of the adipose tissue mass around the rib cage, abdomen, and visceral cavity. This additional mass alters the balance of the inflationary and deflationary pressures on the lungs, ultimately leading to a reduction in the functional residual capacity (FRC) [5,6,7]. Furthermore, as the body mass index (BMI) increases in patients with asthma, a worsening of the airway obstruction is observed, as measured by spirometric parameters such as the forced expiratory volume in 1 s (FEV_1_) and forced vital capacity (FVC) [8,9,10,11].

Our previous research identified a critical limitation in exploring the role of obesity in asthma: the singular reliance on the BMI as the sole measure of obesity. The BMI, a commonly used metric calculated by dividing weight by height squared (kg/m^2^), serves as a general indicator of nutritional status [12]. However, despite its association with the fat mass (FM) and percentage of body fat (PBF), the BMI falls short as a comprehensive proxy for fat distribution and composition [12]. The sensitivity and specificity of the BMI in detecting individuals with excessive PBFs are limited [12,13,14]. Moreover, the BMI fails to distinguish between muscle and fat tissue [12,15], offering an incomplete picture of a person’s body composition. Obesity results from an imbalance between energy intake and expenditure, and a given BMI may represent significantly different metabolic and energetic profiles [16]. Therefore, relying solely on the BMI to assess obesity and its metabolic impact in patients with asthma is insufficient. A more comprehensive approach, incorporating additional metrics, such as body composition analysis, is necessary to accurately assess the role of obesity in asthma and develop targeted therapeutic strategies.

The basal metabolic rate (BMR)—the energy expended by the body at rest to maintain vital functions—has traditionally been a key metric in assessing the metabolic rate and energy expenditure in obesity [17]. Understanding the contribution of the individual BMR to daily energy expenditure is crucial for developing and implementing weight management interventions in patients with obese asthma [17]. Typically, the BMR is estimated using prediction equations that consider various factors, such as age [18,19,20,21,22,23,24,25], gender, race [26], and body composition (including fat mass (FM), muscle mass, and fat-free mass (FFM)) [19]. Among these variables, FFM and muscle mass are the primary determinants of the BMR. Previous research has shown a positive correlation between the BMR and lung function in healthy individuals [27]. Moreover, individuals with asthma tend to have higher BMRs compared to healthy people [28]. However, despite this knowledge, it remains unclear whether the BMR and its associated body composition factors, particularly muscle mass, play a significant role in the airway obstruction in patients with obese asthma. 

In this prospective cohort study, we aimed to address the limitations of relying solely on the BMI to define obesity in individuals with asthma. To achieve this, we employed a multifaceted approach that incorporated the BMI, PBF, and waist circumference (WC) as the metrics to define obese asthma. Crucially, we performed body composition analysis (BCA) on the patients, which allowed us to assess their BMRs, muscle mass, adiposity, and fat-free mass (FFM). This comprehensive analysis enabled us to explore the intricate relationships between the BMR and muscle mass with obese asthma. Furthermore, we delved into the potential mediating roles of the BMR and muscle mass in the association between obesity and the lung function. By doing so, we hoped to gain a deeper understanding of the mechanisms underlying the relationship between obesity and asthma. Lastly, we investigated the longitudinal relationship between the BMR and muscle mass with the lung function over the following year. This allowed us to assess the prognostic values of these metrics in patients with obese asthma. By incorporating multiple metrics and exploring their interactions, we hope to provide valuable insights that can inform the development of nutrition-related multidimensional assessments and interventions for patients with obese asthma.

## 2. Methods 

### 2.1. Study Design and Participants

This prospective real-world cohort study ran from March 2016 to January 2022, with a 12-month follow-up period. Eligible participants were adults (aged 18 years and above) who had been diagnosed with stable asthma and were receiving optimal treatment according to the Global Initiative for Asthma (GINA) criteria at West China Hospital in China [2]. Stable asthma was defined as the absence of respiratory tract infection, exacerbation, or the use of systemic corticosteroids in the preceding four weeks. Patients with other conditions known to affect the BMR, such as hypoadrenocorticism, nephrotic syndrome, pathological starvation, diabetes, erythrocytosis, leukemia, cardiac disease with respiratory distress, and thyroid pathology (e.g., hypothyroidism, hyperthyroidism) indicated by thyroid function tests were excluded from the study. Additional exclusion criteria included an inability to understand the questionnaires and perform spirometry or sputum induction, as well as pregnancy and breastfeeding. Consistent with a real-world study design, patient treatment decisions were guided by the GINA recommendations [2]. This involved a continuous cycle of assessment, treatment, and review, with adjustments made to step up or step down treatments as necessary. The study protocol was approved by the Institutional Review Board (IRB) at West China Hospital, Sichuan University (Chengdu, China) (No. 2014-30). Prior to participation, all individuals provided written informed consent. The study was also registered with the China Clinical Trial Registry (ChiCTR-OOC-16009529; http://www.chictr.org.cn (accessed on 7 April 2024)).

By examining the associations between the BMR, body composition, and lung function in patients with asthma, this study aims to contribute to a more comprehensive understanding of the pathophysiology of asthma in patients with obesity.

### 2.2. Multidimensional Clinical Assessment and Data Collection

We conducted multidimensional data collection, utilizing a standardized case report form to gather information on demographics and clinical characteristics. This encompassed detailed assessments that encompassed anthropometric measurements and body composition analysis, BMRs, muscle mass, medication histories, asthma control assessments, quality-of-life surveys, comorbidity screenings, spirometry tests, fractional exhaled nitric oxide (F_E_NO) measurements, atopy and skin-prick tests, sputum induction procedures, peripheral blood collections, and asthma exacerbation detections. 

### 2.3. Definition of Obesity

The BMI, widely regarded as the standard metric for defining general obesity, has been the focal point of many obesity studies. However, exploring alternative adiposity indicators like WC, a measure of abdominal obesity, or PBF (PBF-defined obesity) could hold significant clinical value and enhance our comprehension of the intricate relationship between obesity and asthma [29]. Therefore, in this study, we employed three adiposity measures—BMI, WC, and PBF—to define obesity based on WHO recommendations. The BMI was calculated by dividing an individual’s body weight (kg) by the square of their height (m^2^). A BMI exceeding 25 kg/m^2^ was considered obese for both Asian men and women [30]. WC (cm) was measured at the level of the umbilicus using an inelastic tape measure. A WC greater than 90 cm for men or 80 cm for women was classified as abdominal obesity [31]. Furthermore, drawing from previous studies and the WHO Technical Report, we adopted a PBF threshold of 25% for men and 30% for women as the cutoff for defining PBF-defined obesity [29,32,33,34]. A diagnosis of obesity in our study was made if any one of these three criteria was met. Based on this assessment, patients were categorized into either the obese asthma group or the non-obese asthma group at the start of the study. This comprehensive approach allowed us to capture a broader spectrum of obesity phenotypes and their potential impact on asthma outcomes.

### 2.4. Body Composition and BMR Measurements

The body composition and BMR were measured using indirect calorimetry under strictly standardized conditions. These measurements were conducted early in the morning (08:00 a.m.), following a minimum of 10–12 h of fasting and at least 30 min of rest. The environment was maintained in absolute silence and at a thermoneutral temperature, with the room temperature set at 25 °C to ensure optimal conditions for accurate measurements [35]. To assess the body composition, including the FM (kg), PBF (%), visceral fat area (VFA) (cm^2^), skeletal muscle mass (SMM) (kg), and appendicular lean mass (ALM) (kg), we utilized a multifrequency bioimpedance analysis (BIA) with the InBody S10 analyzer (Body Composition Analyzer; Biospace Co., Ltd., Seoul, Republic of Korea). This device employs six different frequency impedance measurements (1, 5, 50, 250, 500, and 1000 kHz) and three frequencies of phase angle measurement (5, 50, and 250 kHz) across five segments of the body (right arm, left arm, trunk, right leg, and left leg), providing a comprehensive and accurate assessment of the body composition. The BIA measurements were performed by a trained nutritionist from our research group, who followed the InBody S10 user’s manual and adhered to the recommended guidelines for the clinical application of bioelectrical impedance analysis [36]. Prior to the measurements, patients were instructed to fast overnight, empty their bladders, and wear light indoor clothing. They were then positioned in a standing posture. 

The ALM was calculated as the sum of the muscle mass in the arms and legs [37]. Additionally, the skeletal muscle mass index (SMI) (kg/m^2^) was derived by dividing the ALM by the square of the patient’s height [38]. The BMR was determined using a validated formula derived from BIA, specifically the Cunningham formula, which has been confirmed through comparisons with indirect calorimetry measurements. To further normalize the BMR, we divided it by both height squared (cm^2^) and the BMI, eliminating the confounding effects of height and weight. While dual-energy X-ray absorptiometry (DXA) is recognized as the gold standard for body composition measurement, BIA has been shown to exhibit a strong correlation with DXA, making it a reliable and practical alternative for assessing body compositions in research settings [39,40,41].

### 2.5. Definition of Low Muscle Mass

Currently, there is no consensus definition on low muscle mass, leading to a diversity of proposed criteria and definitions within the field. In our study, we employed several such definitions to assess low muscle mass, including the European Working Group on Sarcopenia in Older People (EWGSOP1,2) [42,43], the Asian Working Group for Sarcopenia (AWGS) [44], the International Working Group on Sarcopenia (IWGS) [45], and the Foundation for the National Institutes of Health Biomarkers Consortium Sarcopenia Project (FNIH) [46].

### 2.6. Lung Function and F_E_NO

All subjects participating in the study were instructed to refrain from using long-acting β_2_-agonists or anticholinergics for at least 24 h prior to their attendance, and to avoid short-acting β_2_-agonists for 12 h or more [47]. In accordance with the standards set by the American Thoracic Society and the European Respiratory Society (ATS/ERS), we utilized a standardized spirometer (CPES/D USB, MedGraphics, Saint Paul, MN, USA) to measure the FEV_1_ and FVC before and 15 min after administering 400 μg of salbutamol (GSK, Burgos, Spain) through a metered-dose inhaler and spacer (150 mL, Vanbo Technology Corp., Shanghai, China) [47]. To ensure accuracy, all subjects were required to perform at least three acceptable and reproducible maneuvers, and the largest FEV_1_ and FVC values were used for analysis. The predicted FEV_1_ and FVC were calculated based on data from the Chinese population [48]. Additionally, the F_E_NO levels were measured using a NIOX analyzer (Aerocrine, Solna, Sweden), adhering to the ATS/ERS recommendations [49,50].

### 2.7. Asthma Control, Quality of Life, and Exacerbation

Asthma control was evaluated using the Asthma Control Questionnaire (ACQ) [51,52], a six-item survey that delves into asthma symptoms, activity limitations, and the frequency of rescue medication usage. The Asthma Quality-of-Life Questionnaire (AQLQ), encompassing 32 questions, provides a thorough assessment of the individual’s quality of life, encompassing activity limitations, asthma symptoms, emotional distress, and environmental stimuli [52,53]. Both questionnaires have established minimal clinically important differences (MCIDs) of 0.5. Notably, these questionnaires have been validated for use in the Chinese population [54,55].

Asthma exacerbation (AE) was defined in accordance with the guidelines by the American Thoracic Society and the European Respiratory Society (ATS/ERS). Specifically, severe AE (SAE) was characterized by the need for systemic corticosteroid treatment lasting at least three days, or an asthma-related hospitalization or emergency department visit resulting in the administration of systemic corticosteroids for a minimum of three days. To maintain consistency in the assessments, any courses of corticosteroids separated by one week or more were deemed separate SAE events [50,56].

### 2.8. Peripheral Blood and Sputum Induction 

Peripheral venous blood samples were collected from fasting individuals into vacuum tubes or tubes containing ethylenediaminetetraacetic acid (EDTA). These samples were then processed (the Sysmex XN-9000 hematology analyzer, Sysmex Corporation, Kobe, Japan) to obtain differential white blood cell counts. Serum total IgE levels were measured (Beckman Image 800 immunoassay analyzer, Beckman Coulter Inc., Brea, CA, USA), with a minimum detectable level of 5.0 IU/mL [52]. 

Sputum samples were also collected and processed using a standardized method described in our previous studies [57,58,59]. This involved inducing sputum production using 0.9% saline, selecting mucus plugs from the saliva, and dispersing them with dithiothreitol. The supernatant of the processed sputum was aspirated and stored at −80 °C for subsequent analysis. Total and differential cell counts were determined using CytoPro 7620 centrifugation-smear (Wescor, Inc., Logan, UT, USA) [52]. 

### 2.9. Statistical Analysis

Categorical variables were summarized as frequencies and proportions, while continuous variables were tested for Gaussian distribution and expressed as means and standard deviations or medians and interquartile ranges. When possible, all continuous data were transformed into a normal distribution. The difference between cohorts for each variable was evaluated using the Student *t*-test or Mann–Whitney U test for the continuous variables and the chi-square test for the categorical variables, as appropriate. The area under the receiver operating characteristic (ROC) curve was used to determine the appropriate cutoff values of the BMR, ALM, and SMM for detecting obese asthma. Model 1 was an unadjusted model. Model 2 was adjusted for potential confounders (age and sex). Spearman correlation analysis was employed to investigate the association between the BMR and muscle mass in patients with obese and non-obese asthma.

Mediation analysis was performed to gain a deeper understanding of the relationship between obesity and lung function (PROCESS Macro [version 3.3] for SPSS [version 23.0, SPSS, Chicago, IL, USA] [50,60]. Using post hoc parallel multiple-mediation models, the study examined whether the association between obesity and the spirometric measures (FEV_1_, FVC, and FEV_1_/FVC) was mediated by the BMR or muscle mass. Adjustments were made for factors such as age, sex, ICS dosage, and SAE in the past year [27,61]. Specifically, the unstandardized path coefficients were the beta (β) coefficients of the multivariable regression models and represent the magnitude and direction of the associations between the variables included in the model. The total unstandardized β effect (path c) represents the effect of obesity on measurements of the lung function when no other mediators were included in the model, while the direct unstandardized β effect (path c’) represents the effect of obesity on measurements of the lung function when mediators were included. Finally, indirect effects (path a_1_b_1_) represent the effect of obesity on measurements of the lung function through the BMR or muscle mass, respectively. If the indirect-effect path is statistically significant, it can be concluded that mediation has occurred. The significance of the indirect effects was tested by bootstrapped 95% confidence intervals (CIs). PROCESS Macro produces bootstrap estimates and bias-corrected 95% confidence intervals (CIs) for indirect effects, and a 95% CI that does not cross zero indicates a statistically significant indirect effect. For the other tests, two-tailed *p*-values < 0.05 were considered statistically significant. Measurements of the lung function between groups at each visit within the 12-month follow-up were compared using analysis of covariance (ANCOVA) adjusted for age. Statistical analysis was performed using SPSS (Version 23.0; IBM Corp., Armonk, NY, USA).

## 3. Results

### 3.1. Subject Characteristics

A total of 598 patients were included in the study (obese asthma, n = 282; non-obese asthma, n = 316) (Figure 1). Table 1 lists the demographic and clinical characteristics of the patients grouped by their obesity status. Obese asthma was more likely in elderly patients (48.0 [41.0, 60.0] vs. 40.0 [31.0, 48.0] years; *p* < 0.001) and constituted a greater proportion of the uncontrolled asthma (25.5 vs. 17.1%; *p* = 0.011). Patients with obesity and asthma had greater airway obstruction, assessed by the FEV_1_ (1.99 vs. 2.29 L, *p* < 0.001; 71.1 vs. 75.7%, *p* = 0.018), FVC (3.02 vs. 3.33 L, *p* < 0.001; 89.3 vs. 92.8%, *p* = 0.011), and FEV_1_/FVC (65.5 vs. 68.2%; *p* = 0.012). There were no significant differences in the percentages of sputum inflammatory cells. In the peripheral blood, patients with obesity had increased neutrophils (4.60 vs. 3.24 × 10^9^/L; *p* = 0.001) but reduced eosinophils (3.63 vs. 3.76%; *p* = 0.016) compared with patients with non-obesity (Table 2). Lower levels of IgE (108.5 vs. 152.19 IU/mL; *p* = 0.051) and F_E_NO (35.5 vs. 42.5 ppb; *p* = 0.011) were observed in patients with obese asthma compared with patients with non-obese asthma (Table 2).

### 3.2. Anthropometric, Body Composition, and BMR Characteristics

The patients with obesity had higher BMIs (25.29 vs. 21.05 kg/m^2^; *p* < 0.001), FM (21.02 vs. 13.10 kg; *p* < 0.001), PBFs (32.79 vs. 24.39%; *p* < 0.001), and VFAs (95.85 vs. 55.40 cm^2^; *p* < 0.001) (Table 3). Furthermore, the patients with obesity had increased muscle mass (SMM: 23.53 vs. 22.10 kg, *p* < 0.001; SMM/Height^2^: 9.22 vs. 8.61 kg/m^2^, *p* < 0.001; SMI: 6.98 vs. 6.49 kg/m^2^, *p* < 0.001; ALM: 17.85 vs. 16.68, kg, *p* < 0.001) but decreased ALM/BMI (0.71 vs. 0.79 m^2^, *p* < 0.001) (Table 3). 

In our study, patients with obese asthma presented higher BMRs (1284.27 vs. 1210.08 kcal/d, *p* < 0.001) compared with patients with non-obese asthma (Table 3). The BMR is largely determined by the total lean mass, especially muscle mass [62]. A reduction in lean mass will reduce the BMR. Currently, there is no uniform definition of low muscle mass worldwide. We used five available criteria (EWGSOP, EWGSOP2, FNIH, IWGS, and AWGS) to define low muscle mass (Table 3). In this study, obese patients had significant lower proportions of low muscle mass in terms of the SMI defined by the EWGSOP (16.0 vs. 26.9%, *p* = 0.001), EWGSOP2 (22.0 vs. 44.3%, *p* < 0.001), IWGS (18.1 vs. 32.6%, *p* < 0.001), and AWGS (10.3 vs. 21.8%, *p* = 0.001). However, a higher proportion of low muscle mass in the ALM/BMI defined by the FNIH was observed (16.3 vs. 3.8%, *p* < 0.001) (Table 3).

### 3.3. Roles of BMR and Muscle Mass in Predicting Obese Asthma

ROC curves were used to assess the predictive power of the BMR and muscle mass for identifying obese asthma, with AUCs greater than 0.5. A BMR cutoff of 1209.5 was chosen for identifying obese asthma. The BMR was adjusted for the BMI and height squared (BMR divided by BMI or height squared (cm^2^)) to control for the effects of weight and height. The BMR/BMI cutoff point was 19.897; the BMR/Height^2^ cutoff point was 494.047. The ALM and ALM/Height^2^ cutoff points were 15.59 and 6.361, respectively. The ALM/BMI cutoff point was 0.638. The SMM and SMM/Height^2^ cutoff points were 20.85 and 8.165, respectively.

The cutoff values were used for the logistic regression analysis (adjusted for age and sex) to explore the association of the BMR and muscle mass with obese asthma (Figure 2). Patients with higher BMRs (Model 1: OR = 1.889; 95% CI, 1.362, 2.621; Model 2: OR = 3.198; 95% CI, 2.081, 4.915) and higher BMRs/Height^2^ (Model 1: OR = 3.348; 95% CI, 2.392, 4.687; Model 2: OR = 3.924; 95% CI, 2.667, 5.773) had increased associations of obese asthma. For muscle mass, patients with higher SMM (Model 1: OR = 1.780; 95% CI, 1.280, 2.474; Model 2: OR = 3.020; 95% CI, 1.966, 4.640), higher SMM/Height^2^ (Model 1: OR = 3.045; 95% CI, 2.119, 4.377; Model 2: OR = 3.836; 95% CI, 2.522, 5.837), and higher ALM/Height^2^ (Model 1: OR = 2.877; 95% CI,2.053, 4.031; Model 2: OR = 4.942; 95% CI, 3.197, 7.639) had higher risks for obese asthma.

### 3.4. Mediation Analyses of BMR and Muscle Mass in Relationship between Obesity and Lung Function

To further explore the relationships between the BMR and muscle mass with the spirometers, we performed parallel mediation analyses (Figure 3). The models showed that the total effect of obesity (c pathway) on the dependent spirometers (FEV_1_, %FEV_1_, FVC, %FVC, and FEV_1_/FVC) was mediated by the BMR (a_1_b_1_ indirect pathway). Thus, obesity was positively associated with the BMR (path a_1_, β = 75.161, *p* < 0.001). Additionally, the BMR was positively associated with the spirometers (path b_1_, all *p* < 0.05). Bootstrapping analysis revealed significant indirect effects of obesity on the measurements of spirometry through the BMR (all *p* < 0.05). Mediation analyses of the BMR/BMI, BMR/Height^2^, ALM/BMI, ALM/Height^2^, and SMM/Height^2^ in the relationship between obesity and the spirometers are shown in Appendix A.

### 3.5. BMR and Muscle Mass Associated with Future Lung Function

The spirometry was conducted at baseline, the first month, the third month, the sixth month, and the twelfth month. A total of 542 subjects with asthma completed the 12-month follow-up (Figure 1). The differences in the spirometers in patients with different BMR and muscle mass levels (by the cutoff values shown above) were compared (Figure 4 and Appendix A). An age–sex-adjusted ANOVA performed on the BMR and muscle mass revealed that, from the baseline to the twelfth month at the end of the follow-up, patients with higher BMRs (all *p* < 0.05), BMRs/Height^2^ (all *p* < 0.05), SMM (all *p* < 0.05), SMM/Height^2^ (all *p* < 0.05), ALM (all *p* < 0.05), and ALM/BMIs (all *p* < 0.05) had consistently increased FEV_1_ values compared with patients with lower BMRs (Figure 4). Similar results were also found for the spirometers, including the %FEV_1_ (Appendix A), FVC (Appendix A), and %FVC (Appendix A).

## 4. Discussion

To the best of our knowledge, our study is the first to explore the BMR in obese asthma and to further explore the relationship between the BMR and muscle mass with the future lung function in patients with asthma. Our study redefined obese asthma through the combined use of three indicators: the BMI, PBF, and WC. Our study found that patients with obese asthma had higher BMRs and higher muscle mass compared to patients with non-obese asthma. Moreover, patients with obese asthma had lower proportions of low muscle mass. Furthermore, we found that higher BMRs and higher muscle mass were associated with obese asthma, which suggested that some patients with obese asthma do not only have higher fat mass but also higher muscle mass, presenting higher BMRs. Mediation analysis further indicated that obesity was directly and significantly correlated with the lung function in the patients with asthma, partially mediated by the BMR or muscle mass. At the next 12-month follow-up, patients with higher BMRs or muscle mass had significantly better lung function.

To date, the GINA has not provided clear recommendations on the assessment criteria or tools for the obese status in asthma. The diagnosis of obesity in asthma is often based on the BMI, an inaccurate measure of the body fat content. Although vast evidence of the adverse effects of obesity on diseases exists, some studies have concluded that obesity improves disease and promotes survival for some diseases [63,64,65]. For instance, Nadi et al. observed a negative association between the BMI and asthma severity [66]. These controversial conclusions can be attributed to the deficiency of the BMI: it does not take into account the muscle mass (lean tissue), body fat content, overall body composition, and racial and sex differences [67,68]. Describing obesity by the BMI can result in an inaccurate assessment of the adiposity. In contrast, adiposity measurements such as the waist: WC or hip ratio (WHR) are strongly associated with disease events. Therefore, this study attempted to perform the multidimensional assessment of the obesity status by three anthropometric indicators and body composition analysis. In our population, if obesity had been defined by the BMI alone, the patients with obesity would have accounted for only 9.9% of the total subjects. However, when the BMI, WC, and PBF were considered, the proportion of patients with obesity increased to 47.2%. We believe that using the BMI alone to define obese asthma may incorrectly estimate the proportion of obesity in the Asian asthma population. Other adiposity measures, alone or in combination with the BMI, may be more appropriate to measure the obesity status, which can better predict asthma-related outcomes and prognoses.

It is intriguing to note that while asthma and obesity exhibit certain overlapping characteristics, there are distinct molecular entities, such as microRNAs, that exert disparate effects on their respective pathologies. For example, a study [69] identified specific microRNAs that exhibit differential expression profiles in asthma and obesity, suggesting unique regulatory roles in the pathogenesis of asthma and obesity. Moreover, our study findings indicate that patients with obese asthma exhibit lower levels of serum IgE and F_E_NO compared to patients with non-obese asthma. This contrasts with the commonly observed higher levels of IgE and FENO in patients with asthma, which are typically associated with increased airway hyper-responsiveness and inflammation [70,71]. The lower levels of IgE and F_E_NO in patients with obese asthma may be attributed to the impact of obesity on the pathophysiology of asthma. Obesity can alter immune responses, potentially leading to a decrease in the production of certain eosinophilic inflammation mediators [72]. Moreover, obesity may result in restricted lung function, which could influence the F_E_NO levels [73]. It is important to note that the reduction in the IgE levels does not necessarily imply milder symptoms in patients with obese asthma, as obesity itself may exacerbate asthma symptoms and the risk of exacerbation [74]. However, the current study’s sample size was limited, and the *p*-value for IgE was close to the conventional threshold of 0.05, indicating that the findings are near the border of statistical significance. Therefore, this observation should be validated in larger sample sizes and with consideration of other potential confounding factors to ensure the stability and replicability of the results. 

The BMR serves as a proxy for the overall metabolic activity of the body and is regarded as a determinant of metabolic health that is independent of body fatness [75,76]. It is characterized as the energy expenditure necessary to preserve the body’s structural and functional equilibrium in a state of rest, fasting, and thermal neutrality. In this study, the patients with obese asthma had higher BMRs than those with non-obese asthma. A significant correlation in the BMR was also found in relation to obesity in patients with asthma. Our results are consistent with a study including a healthy Chinese population, in that the BMRs of adults with obesity were significantly higher than those of adults with normal BMIs [77]. Previous studies have demonstrated that the BMR not only depends on age [78], sex, race/ethnicity [79], height [78], weight [78], BMI [79], WC [78], and FFM [80] but is also influenced by the FM and PBF [78]. These previous findings support our results that patients with obese asthma identified by their BMIs, WCs, and PBFs have higher adiposity and increased muscle mass and BMRs. 

It has been concluded that obesity is associated with a deterioration of the lung function and increased bronchial hyper-responsiveness [81,82]. We also found that obesity causes reductions in the FEV_1_ and FVC. The BMR has been identified as the principal focus of the development and treatment of obesity [18,83]. However, it is still unclear whether the effects of the BMR exist in the relationship between obesity and spirometers in asthma. Our study found that the BMR was positively associated with the spirometers, which implies that the higher the BMR, the higher the FEV_1_, %FEV_1_, FVC, %FVC, and FEV_1_/FVC levels in subjects with asthma. These findings corroborate previous data from the study by Itagi et al. in which an analysis was undertaken on healthy middle-aged individuals [27]. In addition, previous studies have shown that patients with asthma have increased BMRs [28,84]. Interestingly, our study further confirms the important mediation role of a decreased BMR in obesity-induced deterioration in the lung function, which may be one of the key mechanisms for the worsening lung function in obese asthma. Additionally, bootstrapping, a non-parametric resampling technique used in mediation analysis, enabled us to robustly assess the significance of the indirect effects by generating confidence intervals with increased accuracy [85,86,87,88]. To ensure the reliability of our bootstrapping analysis, we addressed its key assumptions, including the independence of the observations and the representativeness of the sample [85,89]. We treated each participant’s measurement as an independent observation, supported by our data collection process, and adhered to comparatively stringent inclusion and exclusion criteria to enhance the sample representativeness. Furthermore, we conducted 5000 bootstrap resamples to ensure stable confidence intervals [90,91,92]. By employing these robust methodological practices, we strengthened the validity of our mediation analysis findings. The bootstrapping analysis provides a precise measure of the uncertainty, enhancing the confidence in our conclusions. The stability of the resampling process and the comparatively large sample size further enhance the reliability of our results.

We conducted additional analyses to further explore the impact of SMM on airflow obstruction in individuals with non-obese and obese asthma. In the participants with non-obesity, those with higher SMM exhibited significantly improved lung function, as indicated by their higher FEV_1_ (2.42 L vs. 2.13 L, *p* = 0.007; 0.93 vs. 0.70, *p* < 0.001) and FVC (3.42 L vs. 2.96 L, *p* < 0.001; 1.06 vs. 0.87, *p* < 0.001) values, compared to those with lower SMM. In the participants with obesity, those with higher SMM demonstrated significantly better lung function, with higher median FEV_1_ (2.38 L vs. 1.47 L, *p* < 0.001; 0.87 vs. 0.63, *p* < 0.001) and FVC (3.44 L vs. 2.41 L, *p* < 0.001; 0.99 vs. 0.87, *p* = 0.006) values, compared to those with lower SMM. Moreover, a significant difference in the FEV_1_/FVC ratio was observed between the groups (0.73 vs. 0.62, *p* < 0.001), with higher SMM associated with a higher ratio. Therefore, regardless of whether they are obese or non-obese, patients with asthma with higher SMM exhibit better pulmonary ventilation function. The significant difference in the FEV_1_/FVC ratio in the participants with obesity is clinically significant, indicating that higher SMM is associated with a lower risk of airflow obstruction in individuals with obese asthma. This suggests that strategies to increase muscle mass may have potential benefits in improving the lung function in individuals with obese asthma. The differential impacts of SMM on the FEV_1_/FVC ratio between the participants with non-obesity and those with obesity highlights the importance of considering the obesity status in mitigating asthma-related airflow limitations.

To further explore the effects of the BMR and muscle mass on the long-term lung function in patients with asthma, we assessed the relationship between the different BMR and muscle mass levels with the lung function in the following year. The results found that the higher the BMR and muscle mass, the better the lung function sustained in the next year. Therefore, we hypothesize that increasing the BMR and muscle mass may be potential therapeutic targets for improving and maintaining long-term lung function for patients with asthma. As mentioned above, previous studies have confirmed that the BMR is elevated in patients with obesity or asthma. Our study further confirms that the BMR is elevated in patients with obese asthma, and that an elevated BMR has a protective effect on the lung function in patients with asthma. In clinical practice, we usually advise patients with obese asthma to lose weight to improve their asthma control and lung function. However, our findings suggest that weight loss alone may lead to deterioration in the lung function because a reduction in the BMI is accompanied by a reduction in the BMR. We therefore recommend that the interventions for obese asthma target both the reduction in adiposity and the maintenance of a high BMR and high muscle mass. For example, exercise training, known for its benefits in improving cardiopulmonary fitness and reducing obesity, emerges as a promising approach to elevate the BMR and SMM. Furthermore, dietary interventions focusing on high protein intake and resistance training have demonstrated effectiveness in increasing the muscle mass and improving the lung function in individuals with obesity [93,94,95]. Additionally, pharmacological interventions, such as β2-agonists and vitamin D supplementation, have been suggested to enhance muscle strength and mitigate obesity-induced asthma exacerbations [93,96]. Future research should investigate the efficacy of these interventions in improving the BMR, SMM, and lung function and reducing the airway obstruction in patients with obese asthma.

There are several limitations to our study. Firstly, the BMI has been used to assess the obesity status in patients with asthma and to define obese asthma for many years. However, in order to assess the obesity status multidimensionally, based on previous nutritional guidelines and studies, although we used multiple indicators simultaneously to define obese asthma, there still may have been an inaccurate assessment of obesity. Future studies could explore multidimensional approaches to assessing the obesity status in patients with asthma. Secondly, our study included only patients from the Chinese population. The results may not be applicable to other asthmatic populations. Studies conducted on healthy Chinese populations have confirmed that an increased BMI is accompanied by not only increased adiposity but also increased muscle mass [97,98]. Additional research is required to ascertain the generalizability of our findings to broader groups of patients with asthma. Thirdly, BIA and dual-energy X-ray absorptiometry (DXA) are regarded as valid techniques for assessing the BC. In this study, the BC was evaluated solely through the use of BIA. Previous reports support the notion that multifrequency BIA yields fat mass and muscle mass measurements that are consistent with those obtained via DXA across diverse cohorts [99,100]. Fourthly, while we acknowledge that a decreased BMR could result from a deteriorating pulmonary function, which, in turn, leads to reduced physical activity and impacts muscle mass and nutritional health, creating a vicious cycle, our longitudinal cohort study was designed to establish the directionality of the relationship, considering the BMR and muscle mass as the exposure variables and the lung function as the dependent variable. 

## 5. Conclusions

In summary, our study establishes a significant link between the BMR and muscle mass and obesity and the lung function in individuals with asthma. The BMR and muscle mass not only play a role in mediating the relationship between obesity and the lung function, but they also determine the long-term lung function. Consequently, it is crucial to incorporate these previously overlooked nutritional indicators, particularly the BMR, into asthma assessments. Our findings pave the way for future clinical studies to adopt a more comprehensive approach to understand the intricate relationship between the nutritional status and lung function in chronic respiratory diseases. We propose that the therapeutic interventions for patients with obese asthma should not solely focus on reducing adiposity but should also prioritize maintaining a high BMR and high muscle mass.

## Figures and Tables

**Figure 1 nutrients-16-01809-f001:**
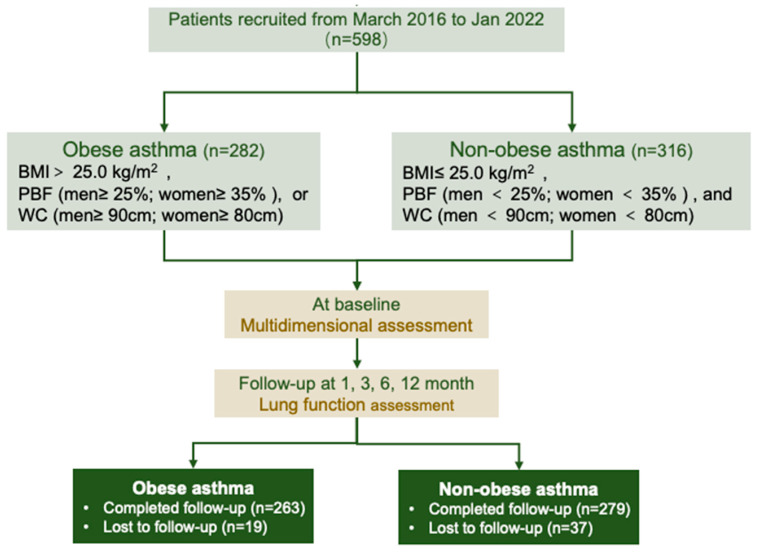
Flowchart of cohort included in the study. BMI: body mass index; PBF: percentage body fat; WC: waist circumference.

**Figure 2 nutrients-16-01809-f002:**
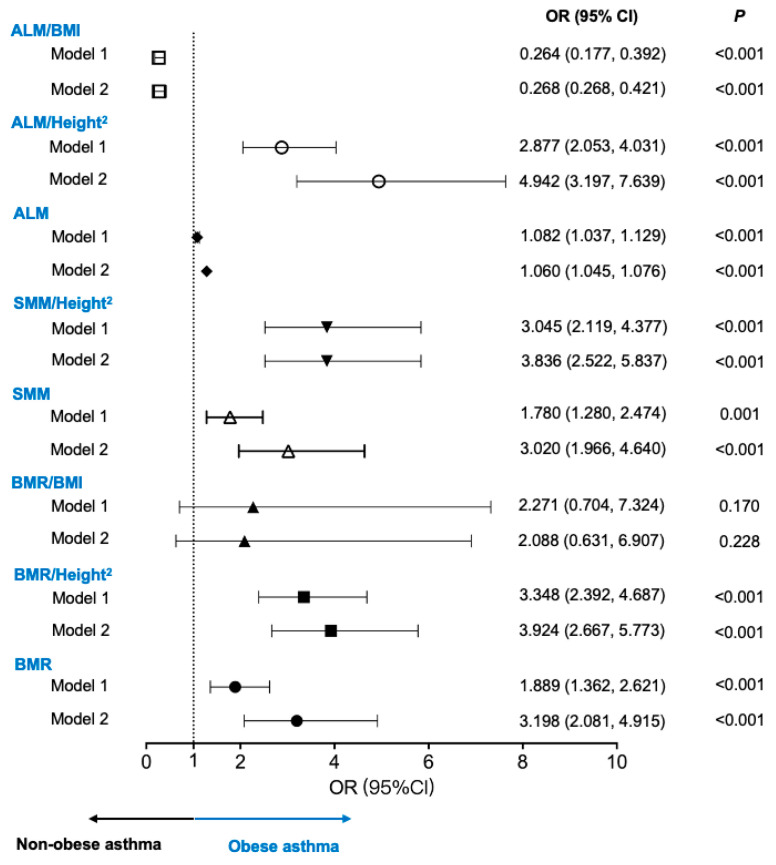
Associations of BMR and muscle mass with obese asthma. Model 1: unadjusted model. Model 2: adjusted for age and sex. ALM: appendicular lean mass; BMI: body mass index; BMR: basal metabolic rate; OR: odds ratio; SMM: skeletal muscle mass.

**Figure 3 nutrients-16-01809-f003:**
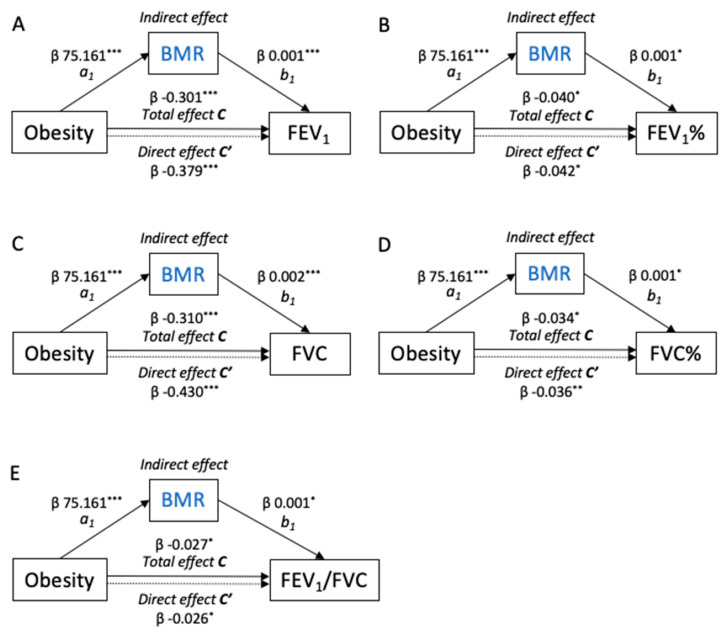
Path model diagrams with the results of the mediation analysis. (**A**) FEV_1_, (**B**) FEV_1_%, (**C**) FVC, (**D**) FVC%, and (**E**) FEV_1_/FVC. Path model showing the effect of obesity on the measurements of the lung function as mediated by the BMR. Total β effect (path c) represents the effect of obesity on the lung function with no mediators in the model. Direct β effect (path c’) represents the effect of obesity on the lung function when the BMR was included in the model. Indirect effects (path a_1_b_1_) represent the effect of obesity on the lung function through the BMR. These models were adjusted for age, sex, ICS dosage, and SAE in the past year. The figures show unstandardized β regression coefficients (* *p* < 0.05, ** *p* < 0.01, *** *p* < 0.001).

**Figure 4 nutrients-16-01809-f004:**
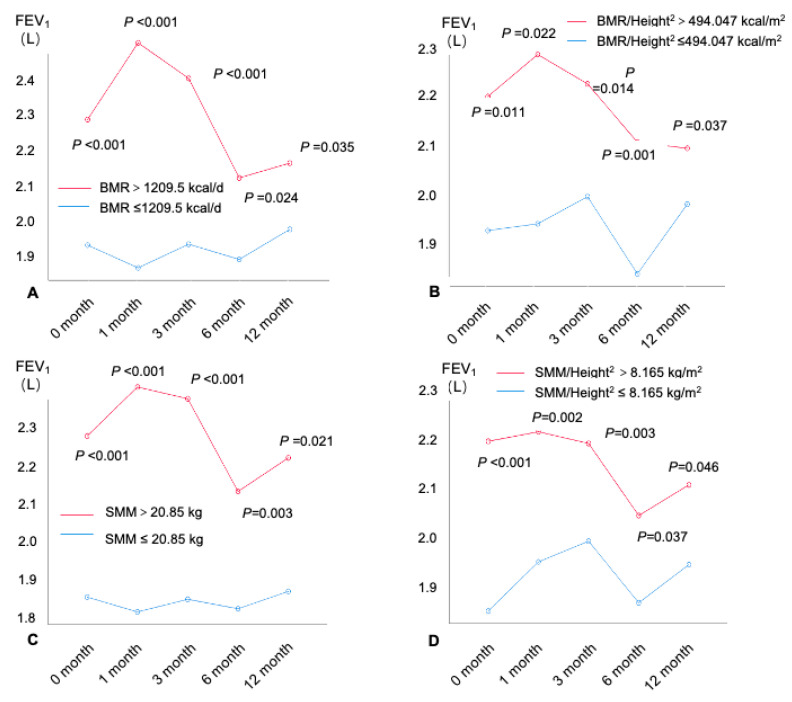
FEV_1_(L) at each visit within the 12-month follow-up by BMR and muscle mass. (**A**) BMR, (**B**) BMR/Height^2^, (**C**) SMM, (**D**) SMM/ Height^2^, (**E**) ALM, (**F**) ALM/Height^2^, and (**G**) ALM/BMI. FEV_1_: forced expiratory volume in 1 s; SMI = ALM/Height^2^. ALM: appendicular lean mass; BMI: body mass index; BMR: basal metabolic rate; SMI: appendicular skeletal muscle index; SMM: skeletal muscle mass.

**Table 1 nutrients-16-01809-t001:** Demographic and clinical characteristics of the included participants with asthma.

Variables	Obese Asthma	Non-Obese Asthma	Total	*t*/*U*/*χ*^2^	*p*-Value
n (%)	282 (47.2)	316 (52.8)	598		
Age, years, median (Q1, Q3)	48.0 (41.0, 60.00)	40.0 (31.0, 48.0)	45.0 (35.0, 55.0)	7.078 *	<0.001
Female, n (%)	183 (64.9)	207 (65.5)	390 (65.2)	0.025	0.875
Atopy, n (%)	99 (35.1)	101 (32.0)	200 (33.4)	0.616	0.433
Asthma duration, years, median (Q1, Q3)	1.0 (0, 4.0)	1.0 (1.0, 6.0)	1.0 (0, 6.0)	−0.372	0.710
Early-onset asthma, n (%)	54 (19.2)	62 (19.6)	116 (19.4)	0.008	0.930
History of family asthma, n (%)	98 (34.8)	109 (34.5)	207 (34.6)	0.001	0.971
Medications					
ICS (BDP equivalent) dose, μg/day, median (Q1, Q3)	400 (400, 1000)	400 (400, 1000)	400 (400, 1000)	−0.042	0.966
ICS/LABA, n (%)	165 (58.5)	182 (57.6)	347 (58.0)	0.065	0.789
Anti-leukotrienes, n (%)	36 (12.8)	49 (15.5)	85 (14.2)	1.068	0.301
Leukotriene, n (%)	98 (34.8)	118 (37.3)	216 (36.1)	0.365	0.546
OCS, n (%)	10 (3.5)	9 (2.8)	19 (3.2)	0.237	0.626
Asthma control					
ACQ-6, median (Q1, Q3)	0.67 (0.17, 1.50)	0.67 (0, 1.34)	0.67 (0, 1.5)	2.107 *	0.036
Uncontrolled asthma (ACQ ≥ 1.5)	72 (25.5)	54 (17.1)	126 (21.1)	6.388	0.011
AQLQ scores, median (Q1, Q3)	5.88 (5.09. 6.32)	5.97 (5.46, 6.50)	5.94 (5.31, 6.41)	−1.587	0.113
SAE in the past year, n (%)	86 (30.5)	78 (24.7)	164 (27.4)	2.530	0.112
Spirometry					
FEV_1_, mean (SD)					
L	1.99 (0.76)	2.29 (0.81)	2.16 (0.80)	−4.647 *	<0.001
%	71.1 (20.1)	75.7 (21.5)	74.4 (20.5)	−2.382	0.018
FVC, mean (SD)					
L	3.02(0.93)	3.33(0.85)	3.20 (0.91)	−4.548	<0.001
%	89.3(17.3)	92.8 (16.4)	91.6 (16.0)	−2.542	0.011
FEV_1_/FVC, %, mean (SD)	65.5 (12.3)	68.2 (14.8)	67.0 (13.1)	−2.509	0.012
Comorbidities, n (%)					
Rhinitis	166 (58.9)	191 (60.4)	357 (59.7)	0.154	0.695
Nasal polyps	26 (9.2)	33 (10.4)	59 (9.9)	0.223	0.637
Bronchiectasis	12 (4.3)	16 (5.1)	28 (4.7)	0.200	0.654
Sleep apnea	3 (1.1)	5 (1.6)	8 (1.3)	0.293	0.589
GERD	21 (7.4)	14 (4.4)	35 (5.9)	2.531	0.112
Diabetes	10 (3.5)	4 (1.3)	14 (2.3)	3.440	0.064
Eczema	46 (16.3)	54 (17.1)	100 (16.7)	0.046	0.830

* Data were normally transformed using *t*-test. ACQ: Asthma Control Questionnaire; AQLQ: Asthma Quality-of-Life Questionnaire; BDP: Beclomethasone dipropionate; FEV_1_: forced expiratory volume in 1 s; FVC: forced vital capacity; GERD: gastroesophageal reflux disease; ICS: inhaled corticosteroid; LABA: long-acting beta-agonist; OCS: oral corticosteroid; Q1: first quartile; Q3: third quartile; SAE: severe asthma exacerbation; SD: standard deviation.

**Table 2 nutrients-16-01809-t002:** Inflammatory characteristics of the included participants with asthma.

Variables	Obese Asthma	Non-Obese Asthma	Total	*t*/*U*/*χ*^2^	*p*-Value
n (%)	282 (47.2)	316 (52.8)	598		
Serum IgE, median (Q1, Q3), IU/mL	108.50 (42.52, 264.25)	152.19 (46.77, 323.99)	104.18 (39.30, 301.62)	−1.948	0.051
FeNO, ppb, median (Q1, Q3)	35.5 (18.0, 65.0)	42.5 (22.0,83.8)	39.0 (20.0, 73.3)	−2.546	0.011
Blood cells, median (Q1, Q3)					
Neutrophils					
%	59.11 (53.09, 64.58)	59.55 (53.19, 65.41)	59.32 (53.12, 64.99)	−1.726	0.084
×10^9^/L	4.60 (3.60, 6.16)	3.24 (2.55, 4.08)	3.41 (2.70, 4.42)	−3.480	0.001
Eosinophils					
%	3.63 (2.02, 5.93)	3.76 (1.99, 6.76)	3.68 (2.0, 6.27)	−2.414	0.016
×10^9^/L	0.22 (0.12, 0.35)	0.25 (0.12,0.41)	0.22 (0.11, 0.36)	−1.157	0.247
≥300 cells/μL, n (%)	102 (36.6)	128 (40.5)	230 (38.7)	0.974	0.324
Eosinophilic asthma *, n (%)	118 (41.8)	143 (45.3)	261 (43.6)	0.726	0.394
Sputum cells, median (Q1, Q3), (n = 353)					
Neutrophils, %	43.50 (17.63, 74.50)	34.5 (13.25, 64.47)	39.0 (15.19, 68.49)	−1.636	0.102
Eosinophils, %	0.25 (0, 2.75)	0.25 (0, 4.25)	0.25 (0, 3.54)	−0.770	0.442
Eosinophils, ≥3%, n (%)	48 (26.1)	51 (30.2)	99 (28.0)	0.730	0.393

* Sputum eosinophil level ≥ 3% or blood eosinophil level ≥ 300 cells/μL. FeNO: fractional exhaled nitric oxide; Q1: first quartile; Q3: third quartile.

**Table 3 nutrients-16-01809-t003:** Anthropometric, BMR, and body composition characteristics.

Variables	Obese Asthma	Non-Obese Asthma	Total	*t*/*χ*^2^	*p*-Value
n (%)	282 (47.2)	316 (52.8)	598		
Anthropometric data					
Weight, kg, mean (SD)	64.25 (11.45)	53.79 (7.72)	58.72 (10.98)	12.944	<0.001
Height, cm, mean (SD)	158.98 (8.34)	159.69 (7.01)	159.35 (7.67)	−1.126	0.261
BMI, kg/m^2^, mean (SD)	25.29 (3.15)	21.05 (2.29)	23.05 (3.45)	18.676	<0.001
≥28, n (%)	59 (20.9)	0 (0)	59 (9.9)	73.35	<0.001
Waist, cm, mean (SD)	89.61 (89.61)	75.06 (6.80)	82.05 (10.42)	22.940	<0.001
Men (n = 208)	93.47 (7.94)	79.64 (7.05)	86.16 (10.18)	12.798	<0.001
≥90 cm, n (%)	62 (62.6)	0 (0)	62 (29.8)	101.899	<0.001
Women (n = 390)	87.62 (7.48)	72.56 (5.18)	79.9 (9.88)	22.28	<0.001
≥80 cm, n (%)	164 (89.6)	0 (0)	164 (42.1)	321.952	<0.001
Hip, cm, mean (SD)	97.66 (6.17)	89.73 (5.70)	93.54 (7.13)	15.837	<0.001
WHR, mean (SD)	0.92 (0.06)	0.84 (0.06)	0.88 (0.07)	15.812	<0.001
Men (n = 208)	0.95 (0.05)	0.88 (0.07)	0.91(0.07)	7.902	<0.001
Women (n = 390)	0.90 (0.06)	0.81 (0.05)	0.86 (0.07)	16.37	<0.001
BMR					
BMR, kcal/d, mean (SD)	1284.27 (235.71)	1210.08 (255.19)	1240.92 (258.49)	3.679	<0.001
BMR/BMI, mean (SD)	51.10 (8.83)	57.88 (12.22)	54.69 (11.26)	−7.836	<0.001
BMR/Height^2^, kcal/m^2^, mean (SD)	505.81(71.09)	473.80 (89.42)	488.89 (82.79)	4.807	<0.001
Body composition					
FM, kg, mean (SD)	21.02 (5.50)	13.10 (3.63)	16.84 (6.07)	20.524	<0.001
PBF, %, mean (SD)	32.79 (5.94)	24.39 (6.16)	28.35 (7.36)	16.916	<0.001
Men (n = 208)	28.36 (4.19)	19.30 (4.38)	23.61 (6.24)	15.198	<0.001
≥25, n (%)	81 (81.8)	0 (0)	81 (38.9)	146.062	<0.001
Women (n = 390)	35.18 (5.34)	27.07 (5.20)	30.89 (6.65)	15.168	<0.001
≥35, n (%)	106 (57.9)	0 (0)	106 (27.2)	164.65	<0.001
VFA, cm^2^, mean (SD)	95.85 (30.27)	55.40 (19.22)	74.48 (32.17)	19.247	<0.001
Men (n = 208)	89.27 (25.67)	50.15 (17.98)	68.77 (29.40)	12.612	<0.001
Women (n = 390)	99.41 (31.99)	58.16 (19.31)	77.52 (33.19)	15.17	<0.001
Muscle mass					
SMM, kg, mean (SD)	23.53 (5.22)	22.10 (4.31)	22.78 (4.81)	3.645	<0.001
SMM/Height^2^, kg/m^2^	9.22 (1.35)	8.61 (1.20)	8.90 (1.31)	5.830	<0.001
Men (n = 208)	10.26 (1.23)	9.64 (0.91)	9.93(1.12)	4.101	<0.001
Women (n = 390)	8.66 (1.05)	8.07 (0.95)	8.34 (1.04)	5.830	<0.001
ALM, kg, mean (SD)	17.85 (4.19)	16.68 (3.50)	17.23 (3.88)	3.666	<0.001
Men (n = 208)	21.78 (3.67)	20.21 (2.70)	20.96 (3.29)	4.471	0.001
Women (n = 390)	15.72 (2.63)	14.83 (2.21)	15.25 (2.45)	3.629	<0.001
SMI (ALM/Height^2^), kg/m^2^, mean (SD)	6.98 (1.11)	6.49 (0.96)	6.72 (1.06)	5.823	<0.001
Men (n = 208)	7.84 (7.84)	7.34 (0.65)	7.58 (0.84)	4.354	<0.001
Women (n = 390)	6.52 (0.89)	6.04 (0.78)	6.28 (0.87)	5.73	<0.001
ALM/BMI, m^2^, mean (SD)	0.71 (0.14)	0.79 (0.15)	0.75 (0.15)	−7.285	<0.001
Men (n = 208)	0.84 (0.12)	0.93 (0.11)	0.89 (0.13)	−5.905	<0.001
Women (n = 390)	0.64 (0.09)	0.72 (0.11)	0.68 (0.11)	−7.988	<0.001
Low muscle mass, n (%)					
EWGSOP					
SMI	45 (16.0)	85 (26.9)	130 (21.7)	10.485	0.001
SMM/Height^2^	14 (5.0)	23 (7.3)	37 (6.2)	1.375	0.241
EWGSOP2					
ALM	3 (1.1)	4 (1.3)	7 (1.2)	0.053	0.819
SMI	62 (22.0)	140 (44.3)	202 (33.8)	33.182	<0.001
FNIH					
ALM/BMI	46 (16.3)	12 (3.8)	58 (9.7)	26.65	<0.001
IWGS					
SMI	51 (18.1)	103 (32.6)	154 (25.8)	16.408	<0.001
SMM/Height^2^	8 (2.8)	13 (4.1)	21(3.5)	0.742	0.389
AWGS					
SMI	29 (10.3)	69 (21.8)	98 (16.4)	14.512	<0.001

SMI = ALM/Height^2^. ALM: appendicular lean mass; AWGS: Asian Working Group for Sarcopenia; BMI: body mass index; BMR: basal metabolic rate; BSA: body surface area; EWGSOP: European Working Group on Sarcopenia in Older People; FM: fat mass; FNIH: Foundation for the National Institutes of Health Biomarkers Consortium Sarcopenia Project; IWGS: International Working Group on Sarcopenia; PBF: percentage body fat; SD: standard deviation; SMI: appendicular skeletal muscle index; SMM: skeletal muscle mass; VFA: visceral fat area; WHR: waist–hip ratio. Definitions of low muscle mass: EWGSOP: ALM/Height^2^: < 7.26 kg/m^2^ (men) < 5.50 kg/m^2^ (women); SMM/Height^2^: 8.87 kg/m^2^ (men) < 6.42 kg/m^2^ (women). EWGSOP2: ALM:20 kg (men) < 15 kg (women); ALM/Height^2^: < 7.00 kg/m^2^ (men) < 6.00 kg/m^2^ (women). FNIH: ALM/BMI: < 0.789 kg/BMI (men) < 0.512 kg/BMI (women). IWGS: ALM/Height^2^: ≤ 7.23 kg/m^2^ (men) ≤ 5.67 kg/m^2^ (women); SMM/Height^2^: ≤ 8.50 kg/m^2^ (men) ≤ 5.75 kg/m^2^ (women). AWGS: ALM/Height^2^: < 7.00 kg/m^2^ (men) < 5.4 kg/m^2^ (women).

## Data Availability

The data that support the findings of this study are available from the corresponding author upon reasonable request. The data are not publicly available due to participant privacy and consent concerns.

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
