# Peer review of "Predictive Roles of Basal Metabolic Rate and Muscle Mass in Lung Function among Patients with Obese Asthma: A Prospective Cohort Study"

_nutrients, 2024, doi:10.3390/nu16121809_

Round 1
Reviewer 1 Report
Comments and Suggestions for Authors
Thank you for submitting your interesting work.
Minor comments as follow:
1. lines 113-116; kindly explain or remove how the following will be achieved "potentially inform the development 115 of more targeted therapeutic strategies."
2. You are writing "The significance of indirect effects ware tested by boot-251 strapped 95% confidence intervals (CI)." Were the assumptions of the bootstrapping met? Can you discuss this in the context of your mediation analysis findings at the discussion section?
3. In figure 2 the arrows start from which number? this is unclear can you start these from the correct number?
4. can you discuss this finding also "Lower levels of IgE (108.5 vs. 152.19 IU/mL; 270 P=0.051) and FENO (35.5 vs. 42.5 ppb; P=0.011) were observed in obese patients compared 271 with non-obese patients with asthma"?
Reviewer 2 Report
Comments and Suggestions for Authors
Zhang et al. focused on hot topic linking to obesity and asthma, which are both widely spread worldwide. They described a higher BMR and muscle mass in obese asthmatics compared to asthmatics patients, without obesity. Interestingly, obesity was directly and significantly correlated with lung function in asthmatics, partially mediated by BMR or muscle mass. Therefore, the authors suggested that asthmatic patients with obesity could need therapeutic treatment focus on promoting a higher BMR and muscle mass over a reduction in adipose tissue.
However, there are some concerns that need to be addressed.
Major Revisions.
The authors claimed that BMR is elevated in patients with obese asthma and that elevated BMR has a protective effect on lung function in asthmatic patients. Indeed, the 12-month follow-up showed how patients with higher BMR or muscle mass had significantly better lung function. Please, clarify why obese asthmatic patients have higher BMR and muscle mass but lower FEV1 at baseline.
Inclusion criteria should include thyroid pathologies which directly influence the BMR.
The authors should specify the intervention nature of study. The NCT reported is linking to Chinese website (China Clinical Trial Registry) and not English one, therefore I can’t check on the website. Further, the authors stated: “All subjects participating in the study were instructed to refrain from using long-acting β2-agonists or anticholinergics for at least 24 hours prior to their attendance, and to avoid short-acting β2-agonists for 12 hours or more”. This sentence confirms the interventional nature of the study. Please, described this in the methods.
It is interesting underline how the relationship between asthma and obesity is very complex. Indeed, papers described how factors over age, sex, BMI, are able to influence the pathogenesis of these diseases. Interestingly, although obesity and asthma share some similarities, there are factors like microaRNAs that influence in a different way both diseases (for instance, PMID: 36143090). Therefore, the authors should stress this aspect.
There are a significant number of patients in therapy with anti-leukotrienes (total= 216). Interestingly, literature data showed that female response better to anti-leukotriene therapy than men (PMID: 31611790). Therefore, sex is a not a negligible confounding factor in the analysis of relationship between obesity and lung function of asthmatic patients. Please, deepen this topic.
Minor Revisions.
Since the sample size is large, it could useful to introduce the number of populations enrolled in the abstract.
Acronyms that appear the first time need to be clarified (line 55: PBF).
Please, introduce medications for comorbidities such as diabetes in the table 1.
It is more intuitive divided the table 1 in two new tables. First for demographics characteristic, second for inflammatory characteristics (for instance FeNO, blood and sputum cells).
In the table 1 appeared Leukotriene as medications but the drugs used for asthmatics therapy are anti-leukotrienes drugs. Please, fix it.
There is a typing error (Line 255, P<.05).
It is more practical list the bibliography with the numbers.
Even if the data are interesting, the image resolution is very low. It is essential to enhance the resolution power of the graphs to promote the understanding of data (for instance: Figure 4).
Comments on the Quality of English Language
The English language quality is overall good and only a general check is required.
Reviewer 3 Report
Comments and Suggestions for Authors
Thei current study aimed to determine the role of BMR and smooth muscle mass in the terms of the association with airflow obstruction in obese patients with asthma.
1. It might be complicaated interpretation that high muscle mass in obese patients causes airflow obstruction because 2. . Does this mean that high muscle worsens airflow limitation in asthma? Metabolic dysfunction may be related to adiposity, which is sometimes accompanied by relatively small ratio of muscle. High ratio of adiposity relative to muscle is well understood to be associated with the distinct pathophysiology of asthma. If so, the result of the current study seems to convey the opposite messages. How do the authors would like the readers to understand the insight on obese asthma based on the current study?
2. Does high muscle cause airflow obstruction in the subanalysis of intracategolies, non-obese and obese ?
3. How intervention would be appropriate to preserve the airflow in obese asthma based on the results of the current study?
4. The presentation of Figure 3 could be improved. What is the direct effect of obesity? Obesity contains many aspects, such as disproportionate adiposity, or high SMM based on the results of the current study?
.
Round 2
Reviewer 2 Report
Comments and Suggestions for Authors
Based on the very detailed authors’ comments, I am pleased to communicate my positive opinion on the quality of the manuscript.
I think the revised version match the criteria of the journal.
Therefore, the manuscript new version could be published.
Author Response
We would like to express our gratitude to the reviewers for their positive assessment of our manuscript. We are pleased to hear that the revisions meet the journal's criteria, and we appreciate the recommendation for publication.
Reviewer 3 Report
Comments and Suggestions for Authors
The proper analysis was conducted to potentially convey the messages with clinical significance.
Thus, in the current version, it may be more clarified that high SMM might be link to the preservation of lung function. Would the authors appreciate high SMM in obese asthmatic patients? If yes, the protective role of SMM could be more emphasised.
Author Response
Thank you for your insightful suggestions. Following the reviewer's suggestion, to clarify that high skeletal muscle mass (SMM) may be linked to the preservation of lung function, we have emphasized the protective role of SMM in our revised manuscript as follows al lines 463-480 as mentioned in Round 1 reply for comment 19:
“We conducted additional analyses to further explore the impact of SMM on airflow obstruction in non-obese and obese individuals with asthma. In non-obese participants, those with higher SMM exhibited significantly improved lung function, as indicated by higher FEV1 (2.42 L vs 2.13 L, p=0.007ï¼›0.93 vs 0.70, p<0.001) and FVC (3.42 L vs 2.96 L, p<0.001; 1.06 vs 0.87, p<0.001) compared to those with lower SMM. In obese participants, those with higher SMM demonstrated significantly better lung function, with higher median FEV1 (2.38 L vs 1.47 L, p<0.001; 0.87 vs 0.63, p<0.001) and FVC (3.44 L vs 2.41 L, p<0.001; 0.99 vs 0.87, p=0.006) compared to those with lower SMM. Moreover, a significant difference in FEV1/FVC ratio was observed between groups (0.73 vs 0.62, p<0.001), with higher SMM associated with a higher ratio. Therefore, regardless of whether they are obese or non-obese, asthma patients with higher SMM exhibit better pulmonary venti-lation function. The significant difference in FEV1/FVC ratio in obese participants is clinically significant, indicating that higher SMM is associated with a lower risk of airflow obstruction in obese individuals with asthma. This suggests that strategies to increase muscle mass may have potential benefits in improving lung function in obese individuals with asthma. The differential impact of SMM on FEV1/FVC ratio between non-obese and obese participants highlights the importance of considering obesity status in mitigating asthma-related airflow limitations.”
